# Teacher-student semi-supervised approach for medical image segmentation

Maria Baldeon Calisto[1][0000−0001−9379−8151]

Departamento de Ingeniería Industrial and Instituto de Innovación en Productividad y Logística CATENA-USFQ, Universidad San Francisco de Quito, Diego de Robles s/n y Vía Interoceánica, Quito, Ecuador 170901
mbaldeonc@usfq.edu.ec,

**Abstract.** Accurate segmentation of anatomical structures is a critical step for medical image analysis. Deep learning architectures have become the state-of-the-art models for automatic medical image segmentation. However, these models require an extensive labelled dataset to achieve a high performance. Given that obtaining annotated medical datasets is very expensive, in this work we present a two-phase teacher-student approach for semi-supervised learning. In phase 1, a three network U-Net ensemble, denominated the teacher, is trained using the labelled dataset. In phase 2, a student U-Net network is trained with the labelled dataset and the unlabelled dataset with pseudo-labels produced with the teacher network. The student network is then used for inference of the testing images. The proposed approach is evaluated on the task of abdominal segmentation from the FLARE2022 challenge, achieving a mean 0.53 dice, 0.57 NSD, and 44.97 prediction time on the validation set.

**Keywords:** Semi-supervised learning · Image Segmentation · Medical Image Analysis.

## 1 Introduction

Accurate segmentation of anatomical structures is a critical step for medical image analysis. Deep learning models have become the de-facto techniques for segmentation tasks given its state-of-the-art performance in various medical datasets [2,1]. However, without an extensive labelled dataset, neural networks can overfit the training data and perform poorly in unseen data points. In the case of medical image segmentation, this is an important limitation because annotating segmentation masks is an expensive and laborious process that requires of an experienced radiologists. Therefore it has become necessary to develop models that leverage unlabeled data information to aid the learning process.

A promising research direction is semi-supervised learning (SSL). SSL models aim to utilize information from unlabelled data to produce predictions that achieve a higher performance than if trained solely with labelled data [16]. Recently, important semi-supervised deep learning models have been proposed for medical image segmentation. Luo et al. [9] developed a dual-task network that

predicts the pixel-wise segmentation map and level set function of the input image. The implemented loss function combines a supervised learning loss with an unsupervised dual-task-consistency loss function. Chen et al. [4] proposed a multi-task attention-based SSL model that combines an autoencoder with a U-Net-like network. The autoencoder is trained to reconstruct synthetic segmentation labels that encourages the segmentation model to learn discriminative latent representations from the unlabelled images. Nevertheless, previous approaches are tested in datasets where the number of classes is small, so when the numbers of classes increases, the complexity and size of the training framework rises importantly.

In this work, we propose a two-phase teacher-student semi-supervised training approach. In phase 1, a three network "teacher" ensemble is trained in a supervised manner using the labelled dataset. In phase 2, a "student" network is trained with the labelled images in a supervised manner, and the unlabelled images using the pseudo-labels provided by the teacher network. The model is tested on the FLARE2022 challenge dataset, that aims to segment 13 abdominal organs. Our model achieves a mean 0.53 dice, 0.57 NSD, and 44.97 prediction time on the validation set. Our experiments demonstrate that using the teacher-student approach increases a 5% the dice metric over using the model trained with only the labelled dataset.

## 2   Method

The proposed method is composed of two phases as displayed in Figure 1. In phase one, three 2D U-Net [13] models are trained with 2D slices in a supervised manner with the labelled training set using a five fold cross validation division scheme. The teacher network is formed by uniting the networks through soft voting. In phase two, a student 2D U-Net is trained in a semi-supervised manner with the labelled and unlabelled images with pseudo-labels provided by the teacher ensemble. Details of each phase are provided next.

### 2.1   Phase One

The training dataset is divided into 5 folds, by assigning 80% of the observations for training and 20% observations for validation. A deeply supervised 2D U-Net, as presented in Figure 2, is trained on each of the folds using the training protocols described in the following section. The 2D U-Net is composed of five down-sampling modules and four up-sampling modules. The modules are comprised of two convolutional blocks, each convolutional block has a $3 \times 3$ convolutional layer, batch normalization layer, and ReLU activation function. The last and second-last up-sampling modules are followed a $1 \times 1$ convolutional layer with a softmax activation function to produce the predicted segmentation. The objective function being minimized during training is a linear combination of the soft dice loss and cross entropy loss as presented in Eq. 1 as it has shown to provide robust results in various medical image segmentation tasks [10].

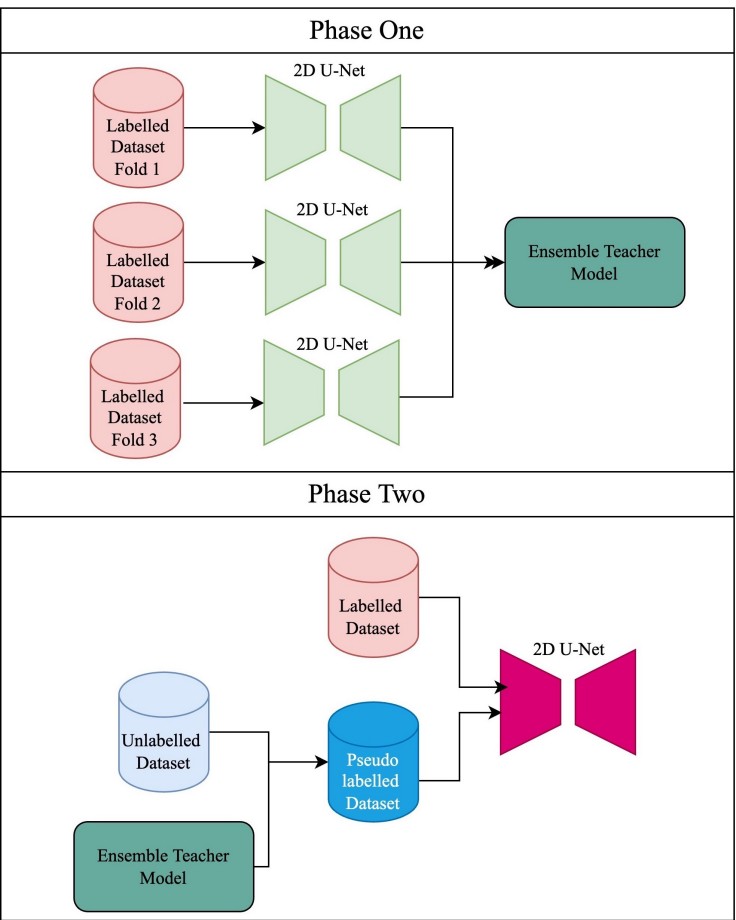

**Fig. 1.** Two-Phase Approach for semi-supervised learning. In Phase 1 a teacher ensemble is trained in a supervised manner. In Phase 2 a student network is trained in an semi-supervised manner using pseudo labels from the teacher.

$$\mathcal{L}_{seg} = \beta \sum_c 1 - \frac{2 \sum_i \widehat{y}_{ic} y_{ic}}{\sum_i \widehat{y}_{ic} + \sum_i y_{ic}} - (1 - \beta) \sum_c \sum_i (y_{ic} log(\widehat{y}_{ic}) \qquad (1)$$

where $y_{ic}$ is the ground-truth label for pixel $i$ in class $c$, and $\widehat{y}_{ic}$ the corresponding predicted probability. $\beta$ is a weight parameter for the dice loss, which we set to 0.65. As previously mentioned, a deep supervised layer with an auxiliary segmentation loss [8] is located in the second-last up-sampling block to aid the model to learn rich hierarchical features. Therefore, the final loss function is comprised of the loss from the main output and the loss from the deep supervised layer with a weight of 0.1.

From the five networks trained, three were selected to form an ensemble as this combination provided the best performance on the challenge's validation set.

### 2.2   Phase Two

In phase two, a 2D U-Net architecture (refer to Figure 2) is trained in a semi-supervised manner to segment the medical images. First, the teacher ensemble network formed in phase one is utilized to produce pseudo-labels for the unlabelled images. During a training iteration, the student 2D U-Net is trained with a batch of 2D labelled images using the same loss function displayed in Eq. 1, and later with a batch of unlabelled images with the psuedo-labels as ground truth with the loss shown in Eq. 2. Here $\tilde{y}_{ic}$ represents the pseudo-label for pixel $i$ in class $c$. An L1 loss between the predicted segmentation and pseudo-label has been added to Eq. 2 as previous work has shown that it incentivizes the segmentations to be consistent [15].

$$\mathcal{L}_{pseudo-seg} = \beta \sum_c 1 - \frac{2 \sum_i \widehat{y}_{ic} \tilde{y}_{ic}}{\sum_i \widehat{y}_{ic} + \sum_i \tilde{y}_{ic}} - (1 - \beta) \sum_c \sum_i (\tilde{y}_{ic} log(\widehat{y}_{ic}) + \\ \sum_c \sum_i ||\widehat{y}_{ic} - \tilde{y}_{ic}|| \qquad (2)$$

The resulting 2D U-Net is used for inference on the validation and testing sets. This trick also allows the single 2D U-Net to learn all the information of the three-network ensemble, performing even better than the ensemble while reducing the size to 1/3.

## 3   Experiments

### 3.1   Dataset and evaluation measures

The FLARE2022 dataset is an extension of the FLARE 2021 [11] with more segmentation targets and more diverse images. The dataset is curated from more than 20 medical groups under the license permission, including MSD [14],

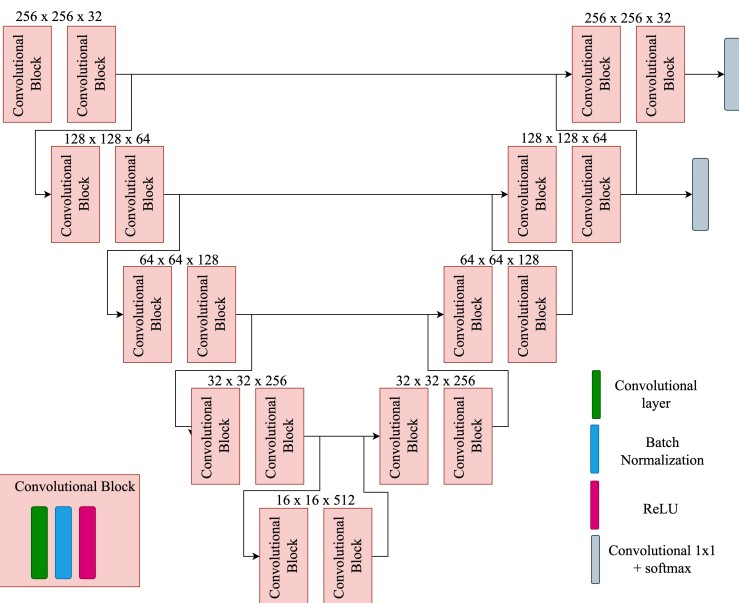

**Fig. 2.** 2D U-Net implemented to segment the abdominal structures. It is composed of five down-sampling modules and four up-sampling modules. The modules are comprised of two convolutional blocks, each convolutional block has a $3 \times 3$ convolutional layer, batch normalization layer, and ReLU activation function.

KiTS [6,7], AbdomenCT-1K [12], and TCIA [5]. The training set includes 50 labelled CT scans with pancreas disease and 2000 unlabelled CT scans with liver, kidney, spleen, or pancreas diseases. The validation set includes 50 CT scans with liver, kidney, spleen, or pancreas diseases. The testing set includes 200 CT scans where 100 cases has liver, kidney, spleen, or pancreas diseases and the other 100 cases has uterine corpus endometrial, urothelial bladder, stomach, sarcomas, or ovarian diseases. All the CT scans only have image information and the center information is not available.

The evaluation measures consist of two accuracy measures: Dice Similarity Coefficient (DSC) and Normalized Surface Dice (NSD), and three running efficiency measures: running time, area under GPU memory-time curve, and area under CPU utilization-time curve. All measures will be used to compute the ranking. Moreover, the GPU memory consumption has a 2 GB tolerance.

### 3.2   Preprocessing

The images have a heterogeneous voxel spacing and shape. Hence, we first resample all images to have a voxel spacing of $1.5mm \times 1.5mm \times 2.5mm$ and set to a fixed size of $256 \times 256 \times 123$ voxels. Moreover, the pixel intensities are clipped to be inside the 3 standard deviations from the mean and rescaled to a [0 , 1] range.

### 3.3   Post-processing

No post-processing operations are applied.

### 3.4   Implementation details

**Environment settings** The environments and requirements are presented in Table 1.

**Table 1.** Environments and requirements.

| | |
|---|---|
| Windows/Ubuntu version | Ubuntu 18.04 |
| CPU | Intel Xeon E5-2698 |
| RAM | 256 GB |
| GPU (number and type) | Four Nvidia V100 32G |
| CUDA version | 10.1.243 |
| Programming language | Python 3.9 |
| Deep learning framework | Pytorch (Torch 1.11, torchvision 0.12.0) |
| Specific dependencies | SimpleITK, nibabel, numpy, albumentation |

**Training protocols** The training protocols for phase one and phase two are presented in Table 2 and Table 3 respectively. On both phases we implement data augmentation on the fly for the labelled dataset using the albumentations library [3]. The operations implemented are horizontal flip, vertical flip, random rotation to a maximum of $+/-$ 90 degrees, elastic transformation, grid distortion, and optical distortion.

**Table 2.** Training protocol phase one.

| | |
|---|---|
| Network initialization | Kaiming Uniform |
| Batch size | 40 |
| Patch size | 256×256 |
| Total epochs | 3000 |
| Optimizer | ADAM ($\beta_1 = 0.5$, $\beta_2 = 0.999$) |
| Initial learning rate (lr) | 0.0002 |
| Lr with polynomial decay | |
| Training time | 216 hours |
| Number of model parameters | 6.98M each 2D U-Net network |
| Number of flops | 9.07G each 2D U-Net network |
| Loss function | Dice loss + Cross-entropy loss |

**Table 3.** Training protocols for phase two

| | |
|---|---|
| Network initialization | Kaiming Uniform |
| Batch size | 20 |
| Patch size | 256×256 |
| Total epochs | 200 |
| Optimizer | ADAM ($\beta_1 = 0.5$, $\beta_2 = 0.999$) |
| Initial learning rate (lr) | $1 \times 10^{-5}$ |
| Lr with polynomial decay | |
| Training time | 48 hours |
| Number of model parameters | 6.98M |
| Number of flops | 9.07G |
| Loss function | Dice loss + Cross-entropy loss |

## 4   Results and discussion

### 4.1   Quantitative results on validation set

The proposed model is tested on the validation set and the evaluation metrics obtained through the challenge's website and displayed in Table 4. We first test the U-Net trained with all the labelled images, which obtained a mean dice of 0.4815. We also test the three U-Net network ensemble obtained in Phase 1, which achieved a 0.4973 mean dice. Finally, we test the proposed semi-supervised phase 1 and phase 2 approach, which increased in approximately 0.05 the mean dice over the single U-Net and 0.02 over the ensemble model. Due to computational limitations, we were not able to train with all the unlabelled images and had to selected a subset of 750 images for the implementation of phase 2. In Table 5, the validation scores for each substructure segmented are shown. The overall result of the model is not high in comparison to the competitor models. This might be caused by the use of 2D CNN instead of a 3D CNN, which does not exploits inter-slice information. Moreover, the testing dataset does not follow the same distribution as the training set. Hence, by not using all the unlabeled images, important information about the testing distribution might be excluded.

**Table 4.** Evaluation metrics on the validation set

| Network | Mean DSC |
|---|---|
| U-Net (supervised training) | 0.4815 |
| Ensemble U-Net (supervised training) | 0.4973 |
| Phase 1 + Phase 2 | 0.5272 |

### 4.2   Qualitative results on validation set

Figure 3 presents examples with good and poor segmentation results. The algorithm performs better in the segmentation of the liver, aorta, and inferior vena cava. Meanwhile, it has problems recognizing and segmenting the duodenum and esophagus. This might be caused by the contrast of the anatomical structures, where the liver and aorta can be differentiated from the other structures while the duodenum has a lower contrast with the surroundings.

### 4.3   Segmentation efficiency results on validation set

The average segmentation efficiency results on the validation set are presented in Table 6.

**Table 5.** Evaluation metrics per substructre on the validation set

| Substructure | Mean DSC | Mean NSD |
|---|---|---|
| Liver | $0.74 \pm 0.25$ | $0.66 \pm 0.24$ |
| Right Kidney | $0.56 \pm 0.38$ | $0.55 \pm 0.35$ |
| Spleen | $0.53 \pm 0.35$ | $0.51 \pm 0.32$ |
| Pancreas | $0.49 \pm 0.30$ | $0.62 \pm 0.30$ |
| Aorta | $0.67 \pm 0.26$ | $0.70 \pm 0.25$ |
| Inferior Vena Cava | $0.59 \pm 0.28$ | $0.58 \pm 0.28$ |
| Right Adrenal Gland | $0.46 \pm 0.31$ | $0.58 \pm 0.36$ |
| Left Adrenal Gland | $0.43 \pm 0.32$ | $0.53 \pm 0.37$ |
| Gallbladder | $0.46 \pm 0.39$ | $0.45 \pm 0.38$ |
| Esophagus | $0.37 \pm 0.37$ | $0.43 \pm 0.42$ |
| Stomach | $0.56 \pm 0.29$ | $0.57 \pm 0.25$ |
| Duodenum | $0.40 \pm 0.27$ | $0.62 \pm 0.28$ |
| Left Kidney | $0.58 \pm 0.37$ | $0.56 \pm 0.35$ |

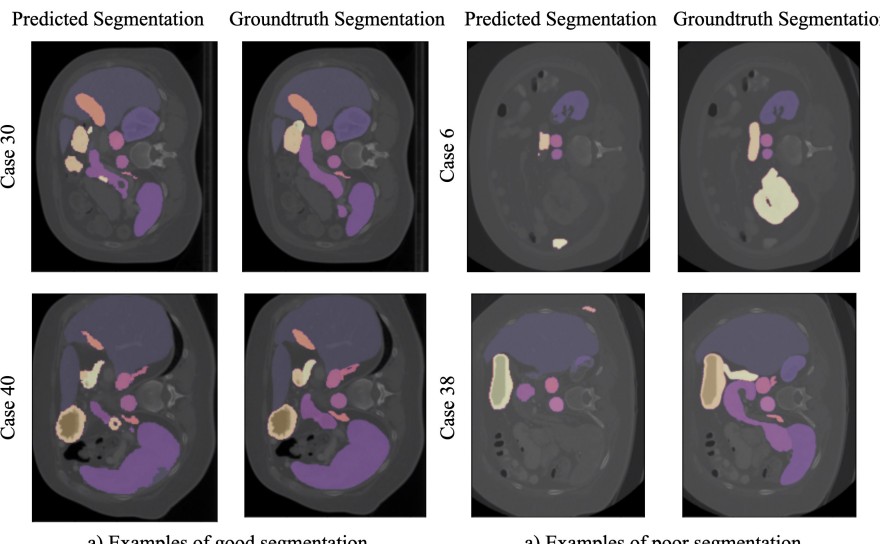

**Fig. 3.** Examples of good and poor performing segmentations. The model produces the best segmentations for the liver, aorta, and inferior vena cava. Meanwhile it has problems recognizing the duodenum and esophagus.

**Table 6.** Average segmentation efficiency metrics on the validation set

| | |
|---|---|
| Time | 44.97 |
| Max GPU Memory | 1405 |
| AUC GPU Time | 49930.08 |
| Max CPU Utilization | 95.10 |
| AUC CPU Time | 785.13 |

### 4.4    Results on final testing set

The model is evaluated on the test through a docker container submission to the challenge. The proposed framework is ranked 30 out of 47 submissions, the evaluation metrics per substructure are presented in Table 7. The model achieves an average 44.39 dice, 46.84 NSD, 44.87 inference time in seconds, 49706 AUC GPU, and 794 AUC CPU.

**Table 7.** Evaluation metrics per substructre on the test set

| Substructure | Mean DSC | Mean NSD |
|---|---|---|
| Liver | $0.62 \pm 0.30$ | $0.52 \pm 0.30$ |
| Right Kidney | $0.56 \pm 0.35$ | $0.51 \pm 0.33$ |
| Spleen | $0.45 \pm 0.37$ | $0.42 \pm 0.36$ |
| Pancreas | $0.37 \pm 0.30$ | $0.48 \pm 0.35$ |
| Aorta | $0.60 \pm 0.24$ | $0.64 \pm 0.24$ |
| Inferior Vena Cava | $0.50 \pm 0.30$ | $0.50 \pm 0.29$ |
| Right Adrenal Gland | $0.42 \pm 0.34$ | $0.52 \pm 0.40$ |
| Left Adrenal Gland | $0.40 \pm 0.33$ | $0.49 \pm 0.39$ |
| Gallbladder | $0.40 \pm 0.40$ | $0.39 \pm 0.40$ |
| Esophagus | $0.24 \pm 0.31$ | $0.28 \pm 0.36$ |
| Stomach | $0.41 \pm 0.29$ | $0.42 \pm 0.28$ |
| Duodenum | $0.28 \pm 0.26$ | $0.43 \pm 0.37$ |
| Left Kidney | $0.51 \pm 0.38$ | $0.48 \pm 0.36$ |

### 4.5    Limitation and future work

A big limitation was the computational memory available. The computing infrastructure is shared between various users, so it was impossible to use all the unlabelled images during training. This was also the reason a 2D network is implemented instead of a 3D network. For future work, we will analyze the confidence of the pseudo labels and implement a GAN to encourage all segmentations to follow a similar distribution.

## 5   Conclusion

In the present work we propose a two-phase semi-supervised learning approach. In the first phase, a three network teacher ensemble is formed by using only the labelled training set. In the second phase, a segmentation network is trained in a semi-supervised scheme using the labelled dataset and unlabelled dataset with pseudo-labels provided by the teacher ensemble. Phase two improved in approximately 0.05 the mean dice from the single U-Net.

**Acknowledgements** The authors of this paper declare that the segmentation method they implemented for participation in the FLARE 2022 challenge has not used any pre-trained models nor additional datasets other than those provided by the organizers.

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
