# OpenReview forum: "Teacher-student semi-supervised approach for medical image segmentation"
_MICCAI.org/2022/Challenge/FLARE_

### Official Review · Reviewer_QD5D · 2022-09-11
**Simple Implementation of Teacher-Student Framework**

**Rating:** 6
**Confidence:** 4

**Review:**

# Summary

The paper first trains 2D U-Net teacher models on different cross-validation sets of the labeled dataset. The best 3 models scoring the highest in the validation set are selected to create an ensemble. The predictions of the ensemble to the unlabelled dataset are used as ground truth for training a student model of the same size.

# Strengths
- Good utilization of teacher-student framework
- Easy to implement

# Weaknesses
- Minor mistakes here and there as if the author didn't review the paper himself
- Explains what was done without the reason behind it

# Details
- Running author is still _F. Author et al._ in every even page from the template
- It is better to change the citation _[6] developed_ to _Luo et al. [6] developed_ for readability
- The paper depicts that the previous approaches dealt only with a small number of classes but provides no evidence of them performing worse in a large number of classes. Also, the paper doesn't introduce a novel way to address this issue.
- The citations like _2D U-Net ronneberger2015u models_ don't provide any information on what the paper was. Please use proper citation style.
- No explanation on why 2D models were used for a 3D dataset
- The equation for cross-entropy seems to be of binary case. Is the segmentation task formulated as a multi-label classification? A pixel can only belong to one class, so the CE loss equation should be used rather than its binary counterpart.
- No explanation for _a deep supervised layer with an auxiliary segmentation loss 12Lee2015 is located in the second-last up-sampling block_
- No explanation on why only 3 models were selected among 5-fold cross-validated models. The obvious way would be to use models trained on all five folds.
- No explanation on what _ỹic_ is. Although it can be inferred to be a pseudo-label, explicitly mentioning it would be clearer for the readers.
- No explanation why an additional MSE loss was used in the second phase for pseudo labels.
- I don't think making the spacing the same is needed because either way there are sampled to the same resolution. If not, please explain it.
- No explanation why no post-processing was applied
- How was the subset of 750 selected from the unlabeled images?
- _Table ??_ should be changed to _Table 5_ ?; the table is hyperlink missing

---

### Official Review · Reviewer_DTSM · 2022-09-14
**Two-phase Teacher-student Semi-supervised Learning**

**Rating:** 6
**Confidence:** 3

**Review:**

Summary:
This work presents a two-phase teacher-student approach for semi-supervised learning. The unlabeled data is used to generate the pseudo-labels to increase the amount of data for training.

Strength:
The unlabeled data is well used.


Weaknesses:
1. No corresponding studies have been conducted to reduce the computational consumption.
2. There are some problems with the layout of the article.

Details:

1. In Section 1, Paragraph 3, Line 2, the description of "teacher" network should match its description in abstract. By the way, please avoid using Arabic numerals at the beginning of acronyms. (e.g. 5-network in abstract)
2. For Fig.1, there is still room to zoom in the image. Horizontal composition is recommended. Besides, the additional explanation of the block in the proposed network is highly expected.
3. Some references do not appear to be cited correctly, please check the code in your latex file.
(e.g. Page 3, Line 2, “ronneberger2015u”; Page 3, Line 19 “12Lee2015” etc.)
4. In Section 2.1, Paragraph 3, what's the link between five-networks and three-networks? And, what rules are the selection process based on? Besides, an equation to represent the final loss function of Phase one is well recommended.
5. For equation 2, the definition of all the variables needs to be added.
6. Lack of qualitative analysis of segmentation results.
7. In Page 7, Line 5, the Table reference failed, please check the code in your latex file.

---

### Official Review · Reviewer_nGiL · 2022-09-15

**Rating:** 7
**Confidence:** 5

**Review:**

Well written paper and clear figures!

A few comments and suggestions:

- The ensemble size is inconsistent in the text, is it 3 or 5?

- It would be nice to have a bit more elaborate figure captions.

- In preprocessing, you set to a fixed size of 256x256x123, does that mean you are cropping images which are larger than this size?

- Table references are not correctly written everywhere

- The main issue that I am missing in the discussion is the big gap in performance between this model (0.53 DSC) and top leaderboard entries (0.88 DSC). A baseline using only the labeled data should get at least 0.75 DSC I think.

---

### Official Review · Reviewer_3SRz · 2022-09-16
**Teacher-student semi-supervised approach for medical image segmentation**

**Rating:** 4
**Confidence:** 3

**Review:**

Strengths: The authors propose a two-phase teacher-student semi-supervised training approach, which increases a 2% the dice metric over using only the labelled data.
Weaknesses:
- Five networks are trained in phase one, but why do you select only three networks to form an ensemble? How do you ensemble the chosen three networks?
- In the proposed method, 2D U-Net is chosen to construct teacher model and student. How do you set the input of 2D U-Net? Do you regard one slice of CT image as a sample?
- It is unclear if you consider the neighbor information between CT slices.
- It is better to use abbreviation of ‘Convolutional Block’ in Fig.2.

---

### Official Review · Reviewer_zate · 2022-09-16
**Basic teacher-student approach using a 2D U-Net**

**Rating:** 3
**Confidence:** 5

**Review:**

This work proposes a simple teacher-student approach for semi-supervised organ segmentation using a 2D U-Net. The final Dice on the test set is 0.5272 using the entire training dataset. There is no novelty in the proposed method, and the results are low compared to other submissions in this validation set. Due to these factors, this work is of limited significance.

---

### Official Review · Reviewer_5JZq · 2022-09-19
**Nice approach to make use of unlabeled data, but important details are missing**

**Rating:** 5
**Confidence:** 4

**Review:**

Summary:
This paper combines a noisy student approach with 2D U-Net training. In the first stage, a 5-fold training strategy is employed, from which 3 folds are selected to make up an ensemble used to create pseudo labels on the unlabeled data. A 2D U-Net is then trained on a combination of the labeled and pseudo-labeled data.

Pros:
- Well-written and straight-forward method description.
- Simple, but effective method to make use of unlabeled data.

Problems:
- References are not shown correctly, requires fixing in latex.
- Figures 1&2 are a bit small and hard to read.
- Missing info on inference requirements in terms of GPU memory, speed, etc. as required by organizer's checklist.
- Missing qualitative comparison as required by organizer's checklist.
- Dice performance below nnU-Net baseline.

---

### Meta-Review · Program_Chairs · 2022-09-28

**Recommendation:** Major Revision
**Confidence:** 5

**Metareview:**

Reviewers raise many concerns and suggestions. Please address all comments in the revised manuscript.